# Performance Test and Parameter Optimization of Trichogramma Delivery System

**DOI:** 10.3390/mi13111996

**Published:** 2022-11-17

**Authors:** Shengzheng Ji, Jinliang Gong, Kai Cui, Yanfei Zhang, Kazi Mostafa

**Affiliations:** 1School of Mechanical Engineering, Shandong University of Technology, Zibo 255000, China; 2Nanjing Dragonfly Smart Agri Research Corp. LTD, Nanjing 210019, China; 3School of Agricultural Engineering and Food Science, Shandong University of Technology, Zibo 255000, China; 4School of Intelligent Manufacturing Ecosystem, Xi’an Jiaotong-Liverpool University, Suzhou 215123, China

**Keywords:** UAV, trichogramma, parameter optimization, feasibility

## Abstract

Trichogramma is a small wasp that is a natural enemy of many agricultural pests. Although Trichogramma can be used in sustainable crop production, conventional methods of delivering Trichogramma to fields are expensive and may cause pollution. In this study, the feasibility of using unmanned aerial vehicles (UAVs) for Trichogramma delivery was investigated. A six-rotor plant protection UAV was equipped with a Trichogramma delivery device, and a Box-Behnken experimental design was carried out with the Trichogramma pills as the test material, the launch height, the launch speed and the launch interval as the experimental factors, and the Trichogramma pills’ landing accuracy as the test index. The data were analyzed by ANOVA using the Design-Expert software, and the influence of each experimental factor on the accuracy of the Trichogramma pills bolus landing was explored through response surface analysis. The regression model between the experimental factors and the experimental indicators was established, and the parameters were optimized based on the response surface method, and the optimal combination parameters were obtained. The ANOVA revealed that the launch height *A* had the greatest effect on the accuracy, followed by launch interval *C* and launch velocity *B*. The results demonstrated that the optimal parameter combination of the Trichogramma delivery system is the launch height of 147.95 cm, the launch speed of 3.7745 m/s, and the launch interval of 2.98 s. At this moment, the accuracy of Trichogramma pills’ bolus landing was the highest, with an accuracy of 93.29%. The average relative error between the experimental value and the optimization result was 1.71%, indicating that the Trichogramma delivery system could meet the requirements of delivery. This study provides theoretical references and technical support for verifying the feasibility of the Trichogramma delivery system.

## 1. Introduction

Pests and diseases substantially affect agricultural production; they are estimated to cause approximately 16% of losses in agriculture yield worldwide [1]. With the popularization of concepts such as green development and healthy living, biological pest control through natural enemies has become an important part of sustainable crop production [2]. Trichogramma is a small wasp that parasitises pest eggs and is commonly used for the biological control of these pests [3,4]. Existing Trichogramma delivery methods include manual delivery, mechanical delivery and drone delivery. Manual delivery is inefficient and labour-intensive, and mechanical delivery methods are fuel-intensive and thus expensive and polluting [5,6]. Multi-rotor plant protection drones have the advantages of simple operation, low operating costs, increased crop yields and improved operational efficiency, which is of great significance in biological control [7,8,9]. The delivery of Trichogramma by multirotor drones is a mainstream idea in pest control; however, a better understanding of the feasibility and accuracy of these system is urgently required.

With the application and promotion of agricultural aviation technology, the use of unmanned aerial vehicles (UAVs) equipped with delivery device can effectively replace manual operation. Numerous experiments regarding Trichogramma delivery with various devices and UAVs have been performed. Tween et al. use fixed-wing aircraft to release insects by natural gravity and suction, avoiding contact with the delivery device and thereby reducing damage to insects. In contrast, the improved delivery system resulted in a more even distribution of insects and a higher survival rate. Because the fixed wing can only fly in a fixed route, it is not flexible and difficult to operate, so it is not suitable for small-area operation [10,11]. Xu et al. [12] designed a Trichogramma delivery system based on the Hex-Rotor, which realized the autonomous delivery function of Trichogramma. However, the influence of the wind flow generated by the rotor on the delivery system is not taken into account, which affects the accuracy of the landing point. Wang et al. [13] designed a delivery device for the harlequin ladybird; however, their device was greatly affected by wind speed and thus also had poor accuracy. Pezzi et al. [14] proposed a backpack wind-borne natural enemy release device primarily comprising a fan, an electromagnet and an air diffuser; wind power was used to achieve the release task. The results indicated that the release mechanism could damage natural enemies and thus must be carefully designed to ensure their survival. Drukker et al. [15] designed a system for dropping insects using aircraft and tested it in the laboratory and in the field. By observing the escape situation, mortality and reproduction rate of insects after release, it is indicated that the system can be used to release insects. The system was evaluated by the efficiency of container opening, the dispersion of containers in the field, and the probability of containers staying in the field, demonstrating the feasibility of aerial release of the delivery system. Pickett et al. [16] used a fixed-wing aircraft as a flight vehicle to release natural enemies with a mechanical device, which caused disadvantages such as poor uniformity and low coverage due to direct contact between the mechanical device and insects. Zhang et al. [17] used the Box-Behnken design (BBD) method to evaluate and optimize the performance of the seed arrangement device. The method could accurately verify the device performance and thus could be used to evaluate Trichogramma delivery systems.

On the basis of the aforementioned research, a Trichogramma delivery system was designed and optimized in this paper. In the BBD tests, the launch height, launch velocity and launch interval were used as the test factors, and the accuracy of the Trichogramma pill landing point was used as the test index. First of all, the data were analyzed by ANOVA using Design-Expert software, and the influence rule of each experimental factor on the dropping accuracy of Trichogramma pills was explored by response surface analysis, and a regression model between experimental factors and experimental indicators was established. Parameters were optimized on the basis of the response surface method, and the best combination of parameters was obtained. Finally, the feasibility of the Trichogramma delivery system was verified by the experiment, and the accuracy of the Trichogramma pill delivery system was improved. The use of drone delivery has improved the efficiency and reduced the cost of the biological control.

## 2. Overall System Structure and Working Principle

### 2.1. Overall Structure

The designed launcher composed a rotary table, a steering gear, a transmitter tube and a motor. Rubber wheel extrusion were used to launch the Trichogramma pills, which effectively avoids the blockage due to friction with mechanical parts. The overall diagram of the Trichogramma delivery system is presented in Figure 1.

### 2.2. Working Principle

The launcher is the core part of the delivery system, and its performance directly affects the operational quality and control effect of Trichogramma. The launcher structure is presented in Figure 2. The function of the diversion device is to facilitate the falling of the Trichogramma pills, the function of the cross turntable is to move the Trichogramma pill, the role of the steering gear is to drive the cross turntable, the role of the motor is to drive the friction wheel, and the role of the fixed plate is to fix the launch tube and the motor. The traditional launchers mostly use a push rod or dial to launch the Trichogramma pills; however, this method is prone to blockage and susceptible to environmental conditions. To achieve delivery, the Trichogramma pills were first loaded into the delivery box, and the UAV flight plan was determined by setting its speed; the flight height and other parameters were set through the cloud platform. When the UAV arrives at the designated delivery point, the UAV monitoring and data forwarding software in the onboard 5G board sends the delivery instruction to a pulse-width modulation (PWM) signal converter, and sends the signal to the steering gear. The steering gear drives the rotary table to release the Trichogramma pills, and the Trichogramma pills fall into the launcher under the action of gravity. Finally, the Trichogramma pills are launched by the friction wheel at high speed, and the launching speed of the launcher is adjusted in real time in accordance with the trajectory of the Trichogramma pills to realize the adaptive control and accurate launching.

## 3. Materials and Methods

### 3.1. Test Materials and Equipment

Commercial Trichogramma pills (approximately 3.4 g each with, diameter of 25 mm) were used. The test equipment adopts the six-rotor plant protection UAV provided by Shandong Folet UAV Manufacturing Co., LTD as the platform, as presented in Figure 3. The UAV model is ZB30L-6, the maximum takeoff weight of 21_kg, the capacity of the delivery box of 10_L, the flight speed of 1–8 m/s, and the effective flight height is of 2–8 m. In order to avoid the influence of weather and other experimental conditions on the experiment, the test is conducted under the conditions of sunny weather, 30 °C temperature, and no wind.

### 3.2. Test Factors and Indicators

#### 3.2.1. The Test Factors

According to previous studies on the Trichogramma delivery system, the launch speed, launch height, and launch interval have a great impact on system performance. Therefore, launch speed, launch height, and launch interval are selected as test factors. The launching speed refers to the speed when the Trichogramma pills leave the delivery system, the launching height refers to the height of the Trichogramma pills from the ground, and the launching interval refers to the delivery time between every two Trichogramma pills. In order to test the performance of the system conveniently and accurately, the corresponding experimental factors were selected according to the research results of previous scholars, combined with the existing test conditions.

(1) Launch height: Downwash airflow will be generated during multirotor UAV operation, and the generated wind field affects the trajectory of released Trichogramma pills. The distribution of this wind field varies with heights [18,19]. Hence, the performance of the Trichogramma delivery system was tested at 100, 130, and 170 cm.

(2) Launch speed: According to the literature [20], launch speed affects displacement distance, so launch speed also has an important influence on the accurate delivery of Trichogramma. The selected launch velocities were 4.0157, 3.8592, and 3.7696 m/s.

(3) Launch interval: According to the research results and test conditions in literature [21], the stability and reliability of the system was tested for launch intervals of 2, 3, and 4 s.

#### 3.2.2. Test Index

No standards for evaluating the performance evaluation of the Trichogramma delivery systems exist. In this paper, the delivery system was evaluated in terms of its landing accuracy. It is based on the distance of the drop point and the effective flight distance of Trichogramma, and the actual dropping distance is measured. Draw a theoretical circle with the landing site as the center and the theoretical proportional distance as the radius. Finally, the number of Trichogramma pills within the theoretical range was counted and the accuracy of the landing point was calculated. The calculation formula of the landing accuracy is:(1)$$R=100\frac{a}{b}$$
where *a* is the number of Trichogramma pills within the circle, and *b* is the total number of Trichogramma pills released.

### 3.3. Experimental Design

The three-factor three-level BBD test method was adopted, and the launch height, launch velocity and launch interval were selected as the test factors. The accuracy of the landing point was taken as the test index. The test factor levels are presented in Table 1.

## 4. Results and Analysis

### 4.1. Test Results

Design-Expert data analysis software was used to process and analyze the test results, and regression analysis and response surface analysis were used to analyze the significant interaction effect between the test factors. [22,23,24] The experimental design and results are presented in Table 2; *A*, *B*, and *C* represent the launch height, launch velocity, and launch interval, respectively, and *R* represents the accuracy of the landing point.

### 4.2. Regression Model Establishment and Result Analysis

The data in Table 2 was analyzed using analysis of variance (ANOVA) [25,26,27] with Design-Expert software; the results are displayed in Table 3. A quadratic response surface regression model of the landing point accuracy *R* on three independent variables: launch height *A*, launch velocity *B*, and launch interval *C* was established. The regression equation was as follows:(2)$$R=89.94+1.10A-0.35B-0.51C-0.19AB-0.32AC-0.63BC-2.14A^{2}-0.035B^{2}-0.85C^{2}$$

To further determine the fitting accuracy of the model and the primary and secondary order of the influence of each factor on the accuracy of the landing point, the analysis of variance and ternary quadratic regression analysis on the accuracy of Table 3 show that there is no significant difference in the regression model. The model correlation coefficient and modified correlation coefficient *S*^2^ and *S*^2^_adj_ were 83.77% and 62.89%, respectively. It shows that the model has a good fit and can be used to analyze and predict the accuracy rate of Trichogramma pills. According to the absolute values of the coefficient of the model, launch height *A* had the greatest effect on deployment accuracy, followed by launch interval *C* and launch velocity *B*. Launching speed and launching height have a significant influence on the accuracy of the landing, and they have significant interaction. The response surfaces and contours of each factor to the accuracy of the landing point are displayed in Figure 4, Figure 5 and Figure 6. According to Figure 4, if the launching speed is fixed, the accuracy of the landing point increases first and then decreases with the increase in the launching height, while the increase in the launching speed has little change in the accuracy of the landing point. As can be observed from Figure 5, the interaction response surface of various factors shows that when the launching speed is fixed, the accuracy of the landing point increases first and then decreases with the increase in the launching interval and the launching height, and the range of change is large. The opening of the response surface is downward and presents a convex shape, indicating that the interaction between the emission interval and the emission height has a significant influence on the response value. As illustrated in Figure 6, when the launching height is fixed, the accuracy of the landing point increases first and then gradually decreases with the increase in the launching interval. With the increase in the launch speed, the accuracy of the landing remains stable. The response surface was flat, indicating that the interaction of the two factors has no significant effect on the accuracy of the landing point.

### 4.3. Parameter Optimization and Experimental Verification

The ANOVA revealed that the launch height *A* had the greatest effect on the accuracy, followed by launch interval *C* and launch velocity *B*. The optimization module in Design-expert software was used to establish a mathematical model for parameter optimization, as follows:(3)$$\left\{ \begin{array}{l} \max R\\ s.t.100\leq A\leq 170\\ \;\;3.7696\leq B\leq 4.0157\\ \;\;2\leq C\leq 4 \end{array}\right.$$

The optimal parameter combination obtained from the module is as follows: launch height of 147.95 cm, launch speed of 3.7745 m/s, and launch interval of 2.98 s.

To verify the accuracy of the model prediction, input the data of the optimal parameter combination into the delivery system. Three drop tests were conducted on the south lawn of the library of Shandong University of Technology, and the test values of the landing accuracy of each drop were recorded and calculated. Finally, the relative error of the landing accuracy is obtained by comparing with the optimized value. In order to avoid the impact of environmental factors on the test, the test was carried out in clear weather at 30 °C with no wind. The UAV was fixed on the bench device (Figure 7).

The parameters of the delivery system were set as follows: launch height of 147.95 cm, launch speed of 3.7745 m/s, and launch interval of 2.98 s. The obtained field verification test results are presented in Table 4. The test results demonstrate that the errors between the test value and the optimal value are 1.50%, 1.61% and 2.03%, respectively. The average relative error between the test value and the optimization result is 1.71%. It shows that the optimal value of the regression equation of each index is close to the test value, and each index can reach better values under the optimal test combination condition.

## 5. Discussion

With the popularization and application of aviation technology in agriculture, the use of unmanned aerial vehicle (UAV) for biological control has become more and more popular, and the problem of traditional manual delivery of insects can be effectively solved by UAV. As the launching device carried by the UAV is still in the test stage, the traditional device is prone to blockage, missed delivery and other phenomena. In turn, the system efficiency and delivery accuracy become worse, and the goal of accurate delivery cannot be achieved. In this study, the blocking problem was effectively solved by using the extrusion launcher, and the optimal parameter combination of the delivery system was calculated by the BBD experimental design method, which verified the feasibility of the Trichogramma delivery system, greatly reduced the cost of the biological control and improved the accuracy of the Trichogramma pill delivery system.

## 6. Conclusions

Design-Expert data processing software and BBD were adopted, and the effects of three factors—launch height (*A*), launch speed (*B*), and launch interval (*C*)—on the accuracy *R* of Trichogramma pills’ deployment were analyzed, and a regression equation with the landing accuracy *R* as the response index was established. An optimization analysis revealed that launch height had the greatest effect on the accuracy, followed by launch interval and launch velocity. The obtained optimal parameter combination was a launch height of 147.95 cm, launch speed of 3.7745 m/s, launch interval of 2.98 s, and the landing accuracy *R* is 93.29%. The test results demonstrate that when the launch height is 147.95 cm, the launch speed is 3.7745 m/s, and the launch interval is 2.98 s, and the landing accuracy *R* is 93.43%, 93.14%, and 93.48%, respectively. The average relative error between the test value and the optimization result is 1.71; thus, the regression model is reliable and can be used for the precise delivery of Trichogramma pills. In the future work, we will consider the influence of UAV wind field on the delivery performance, and consider using the adaptive controller to control the delivery trajectory of Trichogramma pills, to further improve the accuracy of the delivery system.

## Figures and Tables

**Figure 1 micromachines-13-01996-f001:**
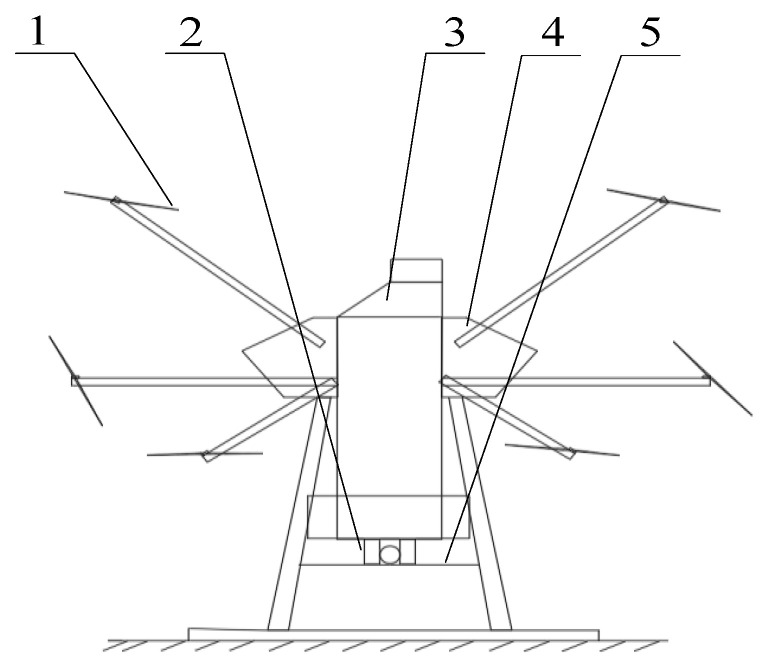
Schematic of the Trichogramma delivery system. 1. Rotor; 2. Release device; 3. Release box; 4. Plant protection UAV; 5. Fixed plate.

**Figure 2 micromachines-13-01996-f002:**
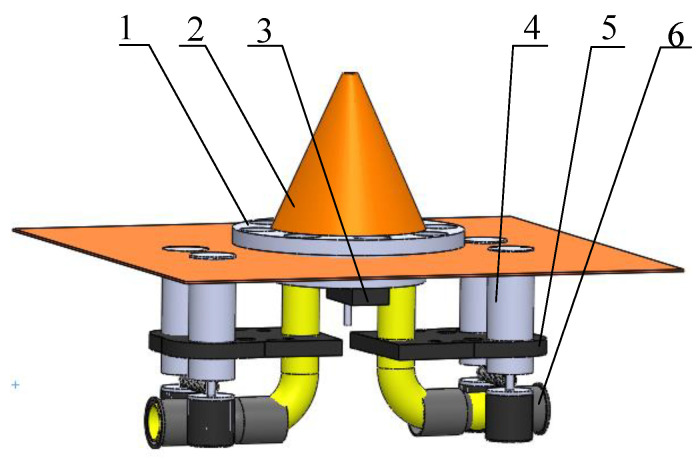
The launcher structure. 1. Cross turntable; 2. Diversion device; 3. Steering gear; 4. Motor; 5. Fixed plate; 6. Launch tube.

**Figure 3 micromachines-13-01996-f003:**
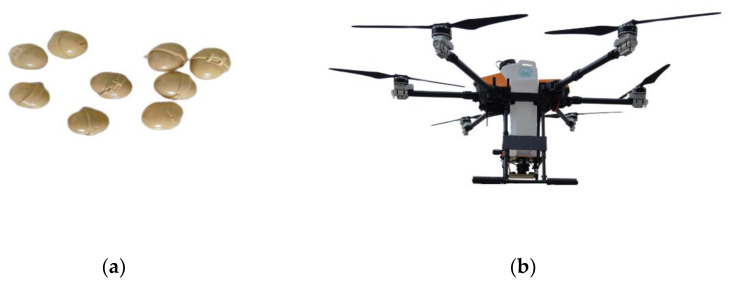
Test materials and equipment. (**a**) Trichogramma pills (**b**) Hexacopter plant protection UAV.

**Figure 4 micromachines-13-01996-f004:**
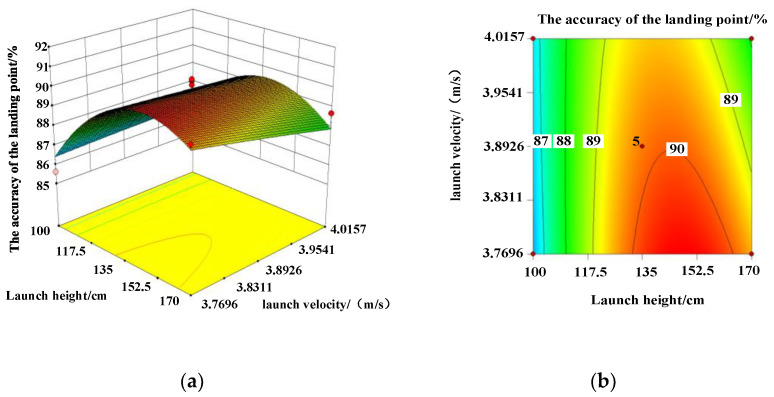
(**a**) Response surface and (**b**) contour plot for the interaction between launch height and launch velocity on landing accuracy. (**a**) Response surface; (**b**) contour lines.

**Figure 5 micromachines-13-01996-f005:**
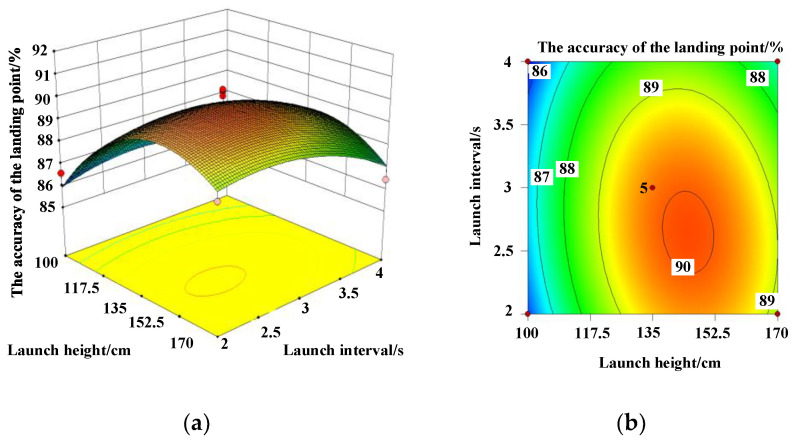
(**a**) Response surface and (**b**) contour plot for the interaction between launch height and launch interval on landing accuracy. (**a**) Response surface; (**b**) contour lines.

**Figure 6 micromachines-13-01996-f006:**
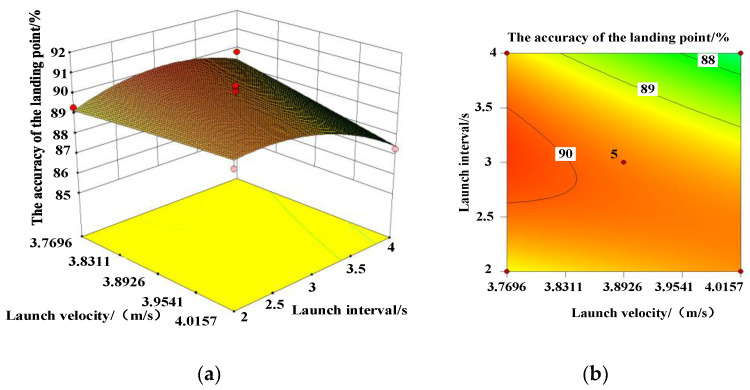
(**a**) Response surface and (**b**) contour plot for the interaction between launch velocity and launch interval on landing accuracy. (**a**) Response surface; (**b**) contour lines.

**Figure 7 micromachines-13-01996-f007:**
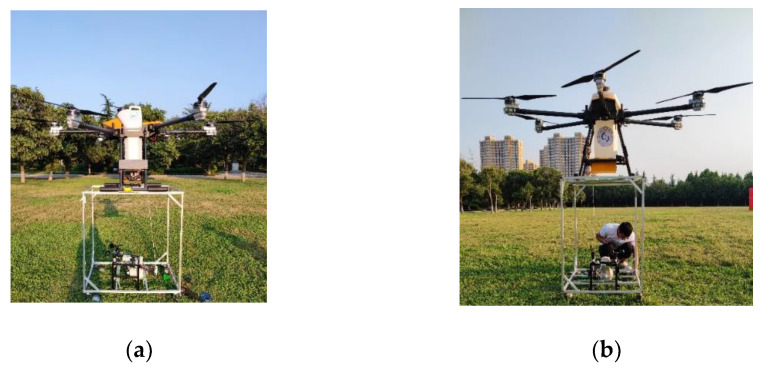
(**a**) Angle 1 and (**b**) angle 2 test on site. (**a**) Angle 1 test on site; (**b**) angle 2 test on site.

**Table 1 micromachines-13-01996-t001:** Table of test factor levels.

Level	Launch Height (m)	Launch Velocity (m/s)	Launch Interval (s)
−1	130	4.0157	2
0	150	3.8592	3
1	170	3.7696	4

**Table 2 micromachines-13-01996-t002:** Experimental design and results.

Test Number	The Test Factors			The Accuracy of the Landing Point *R*(%)
	A	B	C	
1	0	1	−1	89.35
2	1	0	1	86.46
3	−1	−1	0	85.63
4	0	1	1	87.32
5	0	0	0	90.43
6	1	−1	0	89.87
7	0	−1	1	89.81
8	−1	1	0	86.45
9	0	0	0	90.13
10	0	0	0	87.93
11	1	0	−1	88.37
12	0	0	0	90.32
13	0	−1	−1	89.34
14	−1	0	1	85.97
15	1	1	0	88.72
16	−1	0	−1	86.59

**Table 3 micromachines-13-01996-t003:** ANOVA for regression models.

Indicators	Sources of Variance	Sum of Squares	Degrees of Freedom	The *F* Value	The *p* Value	Significant
The accuracy of the landing point *R*(%)	Model	39.10	9	4.01	0.0402	
	*A*	9.64	1	8.90	0.0204	
	*B*	0.99	1	0.91	0.3715	
	*C*	2.09	1	1.93	0.2072	
	*AB*	0.97	1	0.90	0.3753	
	*AC*	0.42	1	0.38	0.5550	
	*BC*	1.56	1	1.44	0.2687	
	$A^{2}$	19.32	1	17.84	0.0039	
	$B^{2}$	5.012 × 10^−3^	1	4.629 × 10^−3^	0.9477	
	$C^{2}$	3.07	1	2.84	0.1358	
	Residual	7.58	7			
	Loss of quasi	2.94	3	0.85	0.5360	Not Significant
	Error	4.64	4			
	Sum	46.68	16			

**Table 4 micromachines-13-01996-t004:** Model optimization and experimental comparison.

Project	The Accuracy of the Landing Point *R*(%)		
	Value of Test	Value of Optimization	Error of Relative
The first time	93.43	93.29	1.50
The second time	93.14	93.29	1.61
The third time	93.48	93.29	2.03
Average value	93.35	93.29	1.71

## Data Availability

The data that support the findings of this study are available from the corresponding author upon reasonable request.

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
