# Peer review of "Performance Test and Parameter Optimization of Trichogramma Delivery System"

_micromachines, 2022, doi:10.3390/mi13111996_

Round 1
Reviewer 1 Report
The research is a new scientific idea, and researchers deserve praise and thanks for this valuable research.
There are some minor notes that I would like to amend:
1/ Please, clarify the image of the drone from several angles.
2/ Please, put Equation 2 inside the text.
3 / Please, put the caption of Figure 5 on the same figure page.
4 / Please, more clarify Figure 7, and from several angles so that the device is optically clearer.
5 / The authors can attach a link to a video containing the stages of experiments conducted.
Author Response
Response to Reviewer 1 Comments
Point 1: Please, clarify the image of the drone from several angles.
Response 1: First, thank you very much for your valuable advice. According to your advice, we have readjusted the resolution of the image to make it clearer, and uploaded the image of the UAV test site from two angles.
Point 2: Please, put Equation 2 inside the text.
Response 2: Thank you very much for your valuable advice, we have made correction according to your comments.
Point 3: Please, put the caption of Figure 5 on the same figure page.
Response 3: Thank you for your suggestion, the caption of Figure 5 is already on the same figure page.
Point 4: Please, more clarify Figure 7, and from several angles so that the device is optically clearer.
Response 4: Thank you for your comments, we have revised the resolution of Figure 7 and shown it from two angles to make the device clearer.
Point 5: The authors can attach a link to a video containing the stages of experiments conducted.
Response 5: We sincerely appreciate your suggestion; however, we are sorry that we cannot provide a video of the experimental stage because there was no video recording during the testing stage; and sincerely thank you again.

Reviewer 2 Report
1. I have many concerns about the style of this paper. I think that if the authors wish this paper is well considered by experts, more attention should be devoted to discuss the application scenario. I suggest to simplify it or better explain with realistic examples.
2. From the Introduction section 1, it is rather unclear what gaps the manuscript attempts to explore and the overall added value of the paper.
3. The authors are suggested to investigate and compare how different algorithms contributed and the scope of improvement. The authors can make a discussion section mentioning the contribution and scope of each included study.
4. In Conclusion, it is suggested to add the validation of your study and discuss the scope of improvement and future scope of this survey.
5. The grammatical errors must be carefully checked and removed from the entire manuscript.
Author Response
Response to Reviewer 2 Comments
Point 1: I have many concerns about the style of this paper. I think that if the authors wish this paper is well considered by experts, more attention should be devoted to discuss the application scenario. I suggest to simplify it or better explain with realistic examples.
Response 1: First, thank you very much for your valuable advice. The style of this paper is mainly based on experiments to verify the feasibility of the performance of the delivery system and to optimize its parameters. As you said, discussing as much as possible the application of the delivery system in the research field is more likely to be considered by experts, and we will discuss the application scenarios in our future work.
Point 2: From the Introduction section 1, it is rather unclear what gaps the manuscript attempts to explore and the overall added value of the paper.
Response 2: First, thank you for your suggestion. Based on your advice, we have revised the introduction to make the overall added value of the manuscript clearer.
Point 3: The authors are suggested to investigate and compare how different algorithms contributed and the scope of improvement. The authors can make a discussion section mentioning the contribution and scope of each included study.
Response 3: Thank you very much for your valuable advice, which are very helpful for our research. We have added a discussion section to the manuscript, investigated and compared the problems of different delivery systems, highlighted the improvement methods and contributions of this study. (lines244-255)
Point 4: In Conclusion, it is suggested to add the validation of your study and discuss the scope of improvement and future scope of this survey.
Response 4: Thank you for your precious comments and advice. We have revised the manuscript according to your advice. Based on your advice, we add validation tests for the study to the conclusion and discuss the future of the study.
Point 5: The grammatical errors must be carefully checked and removed from the entire manuscript.
Response 5: Thank you again for your valuable comments and suggestions, we invited a native English speaker to polish our manuscript, the entire manuscript grammatical errors has been modified.

Reviewer 3 Report
The study of transportation and delivery of biological control has some application value for agricultural control.
The experimental process is relatively rigorous, but some details in the experimental design part are not handled properly:.
(1). In part 2.2, when the working principle is described, the mechanical working principle of the firing device is not clearly described, and the role of each component in the firing process is not clear.
(2). In part 3.2.1, when the test factors and levels were selected, sufficient reasons were not given. For example, the UAV travels at a height of 2-8 m. Why 100, 130 and 170 cm were selected as the factor levels.
(3). In section 3.2.2, when describing the landing accuracy "It is based on the ratio between the distance of the drop point and the effective flight distance of Trichogramma. " should be modified to "It is based on the distance of the drop point and the effective flight distance of Trichogramma." in the following equation In the formula below, the landing accuracy is the ratio of "the number of Trichogramma pills within the theoretical range" to "the total number of Trichogramma pills released", which is easy to be ambiguous.
(4). In addition, it is mentioned in the quotation that the survival rate of Trichogramma plays a very important role in the design of the system. Therefore, the survival rate can be also used as an indicator in the selection of indicators in section 3.2.2 to make the optimization of parameters more practical.
(5). Before part 4 Results and analysis, a control and design of environmental factors should be added, because the experiment is more influenced by environmental factors as mentioned in the citation. For example, the experimental wind speed, weather and other factors.
(6). In part 4, R stands for landing accuracy and R2 stands for correlation coefficient, which is confusing, and it is suggested to change the parameter names.
(7). In part 4.2, Figure 6, there is a mismatch between the conclusion and the icon, from the figure we can see that as the launch interval increases the landing accuracy increases and then decreases or directly decreases, while the text uses "With the increase of the launch interval, the accuracy of With the increase of the launch interval, the accuracy of landing remains stable.
(8). The predicted value of landing accuracy in Table 4 of Section 4.8 is too abrupt and should be mentioned in lines 209 to 212.
(9). The introduction description is relatively simple. The authors were suggested that there should be more research papers in the same field aiming at improving landing point accuracy. Also there should be a corresponding comparison in the research process.
(10). This piece seems more like a report. Certain conclusions were obtained by means of a combination of experiments. Section 2 of the manuscript does not seem to have a positive effect on the presentation of the study in the manuscript. Could this section be replaced with a more detailed presentation of the experimental conditions? Because the study of the manuscript is based on experiments. However, there is less description of the test conditions in the whole text, such as measurement accuracy, test equipment, etc.
(11). Model accuracy may be a more serious problem with this manuscript. The sample size is only 16 groups, while the polynomial model already has 10 parameters. Moreover, the model accuracy is relatively limited in terms of numerical values. Therefore, more combinatorial experiments based on BBD may provide more help to the study. In addition, different modeling approaches may also be able to be added to the study. Of course, this suggestion is only an academic discussion.
Author Response
Response to Reviewer 3 Comments
Point1: In part 2.2, when the working principle is described, the mechanical working principle of the firing device is not clearly described, and the role of each component in the firing process is not clear.
Response 1: First, thank you for your precious comments and advice. Those comments are all valuable and very helpful for revising and improving our paper. Based on your advice, in part 2.2, we redescribed the working principle of the firing device and explained the function of each component. (lines100-103)
Point2: In part 3.2.1, when the test factors and levels were selected, sufficient reasons were not given. For example, the UAV travels at a height of 2-8 m. Why 100, 130 and 170 cm were selected as the factor levels.
Response 2: Thank you for the suggestion. In part 3.2.1, we have added the information required as explained above. (lines134-141)
Point3: In section 3.2.2, when describing the landing accuracy "It is based on the ratio between the distance of the drop point and the effective flight distance of Trichogramma. "should be modified to "It is based on the distance of the drop point and the effective flight distance of Trichogramma." in the following equation In the formula below, the landing accuracy is the ratio of "the number of Trichogramma pills within the theoretical range" to "the total number of Trichogramma pills released", which is easy to be ambiguous.
Response 3: Thank you very much for your valuable advice, which are very helpful for our research. Based on your advice, the statement has been modified. The landing accuracy is redescribed to avoid ambiguity. (lines153-160)
Point4: In addition, it is mentioned in the quotation that the survival rate of Trichogramma plays a very important role in the design of the system. Therefore, the survival rate can be also used as an indicator in the selection of indicators in section 3.2.2 to make the optimization of parameters more practical.
Response 4: Thank you for your precious comments and advice, it is really true as your suggested that the survival rate of Trichogramma plays a very important role. However, the survival rate of Trichogramma is not only related to the parameters of the delivery mechanism, but also related to the environment and Trichogramma itself. Therefore, it should be considered comprehensively, and it can be taken into consideration in future work.
Point5: Before part 4 Results and analysis, a control and design of environmental factors should be added, because the experiment is more influenced by environmental factors as mentioned in the citation. For example, the experimental wind speed, weather and other factors.
Response 5: Thank you for your comments, we have made correction according to your comments. The control and design of environmental factors were added before part 4 results and analysis. (lines126-128)
Point6: In part 4, R stands for landing accuracy and R2 stands for correlation coefficient, which is confusing, and it is suggested to change the parameter names.
Response 6: Thank you for the suggestion, according to your advice, we have changed the names of the parameters; we changed R to S. (line186)
Point7: In part 4.2, Figure 6, there is a mismatch between the conclusion and the icon, from the figure we can see that as the launch interval increases the landing accuracy increases and then decreases or directly decreases, while the text uses "With the increase of the launch interval, the accuracy of With the increase of the launch interval, the accuracy of landing remains stable.
Response 7: In part 4.2, we mistakenly wrote launch speed as launch interval and launch interval as launch speed. We are very sorry for this. Thank you very much for your reminding. This error has been corrected. (lines201-204)
Point8: The predicted value of landing accuracy in Table 4 of Section 4.8 is too abrupt and should be mentioned in lines 209 to 212.
Response 8: Thank you for your comments, according to your suggestions, we have explained the accuracy of the landing accuracy accordingly. (lines229-231)
Point9: The introduction description is relatively simple. The authors were suggested that there should be more research papers in the same field aiming at improving landing point accuracy. Also there should be a corresponding comparison in the research process.
Response 9: Thank you for the suggestion, according to your comments, we have added relevant research in the same field to the introduction and made corresponding comparisons with it. (lines64-72)
Point10: This piece seems more like a report. Certain conclusions were obtained by means of a combination of experiments. Section 2 of the manuscript does not seem to have a positive effect on the presentation of the study in the manuscript. Could this section be replaced with a more detailed presentation of the experimental conditions? Because the study of the manuscript is based on experiments. However, there is less description of the test conditions in the whole text, such as measurement accuracy, test equipment, etc.
Response 10: We sincerely appreciate your suggestion, In the section 2 of the manuscript, in order to make readers more clearly understand the structure and working principle of the whole delivery system, I feel it is necessary to keep this part, and added the function of each part. According to your suggestion, we have added the detailed introduction of test equipment, experimental environment and other conditions. (lines122-128)
Point11: Model accuracy may be a more serious problem with this manuscript. The sample size is only 16 groups, while the polynomial model already has 10 parameters. Moreover, the model accuracy is relatively limited in terms of numerical values. Therefore, more combinatorial experiments based on BBD may provide more help to the study. In addition, different modeling approaches may also be able to be added to the study. Of course, this suggestion is only an academic discussion.
Response 11: Our deepest gratitude goes to you for your careful work and thoughtful suggestions that have helped improve this paper substantially. As you said, more combinatorial experiments based on BBD may provide more help to the research and improve the accuracy of the model. The sample size of these 16 groups is to select the best data from all the data, which I think is enough to explain the accuracy of the model. Of course, we can also choose different modeling methods for research and compare these modeling methods to get the best results. Thank you again for your valuable comments.

Reviewer 4 Report
The topic addressed by the authors is topical. The performance test and the optimisation of the transport system parameters are fulfilled, although it would be interesting to see what happens when there is no calm, although these can be investigated with further measurements. In terms of content, the paper is a coherent, compact work.
Formal comment: the text in Figure 5 is in bold.
Author Response
Response to Reviewer 4 Comments
Point1: The topic addressed by the authors is topical. The performance test and the optimisation of the transport system parameters are fulfilled, although it would be interesting to see what happens when there is no calm, although these can be investigated with further measurements. In terms of content, the paper is a coherent, compact work.
Formal comment: the text in Figure 5 is in bold.
Response 1: Thank you very much for your acknowledgement of this study in terms of content and subject matter, and we highly appreciate your time and consideration. According to your suggestion, we have adjusted the text in Figure 5 in bold.

Reviewer 5 Report
The authors should go through the following review points to improve the quality of the manuscript.
1. Abstract section should be improved.
2. The Introduction section should be improved by maintaining the proper workflow.
3.The authors should add a "Related Works" section and describe minimum 15-20 related works with their identified gaps.
4.The methodology part is not described properly. The authors should describe this section clearly by adding required algorithms and their mathematical implications.
5.The results and analysis section should be improved by adding more graphs, figures, tables, etc. to improve the quality of the manuscript.
6. The conclusion section should be improved.
7.The authors should clearly describes the future scope of this work.
8. The authors should add 5-7 recent relevant references in the References section and cite them properly in the text.
9. The authors should go through the entire manuscript thoroughly and remove all the typo/grammatical errors from it.
Author Response
Response to Reviewer 5 Comments
Point1: Abstract section should be improved.
Response 1: First, thank you very much for your valuable advice, which are very helpful for our research. According to your suggestion, we have modified the grammatical errors and relevant data in the summary, which is consistent with the full text of the manuscript.
Point2: The Introduction section should be improved by maintaining the proper workflow.
Response 2: Thank you very much for your suggestions, we have improved the introduction section to maintaining the workflow of this study.
Point3: The authors should add a "Related Works" section and describe minimum 15-20 related works with their identified gaps.
Response 3: Thank you for your comments; according to your suggestions, we added a discussion section, which described the shortcomings of the previous research in detail and filled the gaps through our study. For example, literature [3-6] mentioned the use of manual delivery, which is inefficient and costly, and this study verified the feasibility of unmanned aircraft delivery, significantly reducing labor costs. The delivery device in the literature [10-15] is prone to blocking during delivery. Our study uses a friction wheel to squeeze and launch the Trichogramma pills, effectively avoiding this phenomenon.
Point4: The methodology part is not described properly. The authors should describe this section clearly by adding required algorithms and their mathematical implications.
Response 4: Thank you for your advice, this study is based on an experiment, and the modeling method used for the three-factor three-level BBD test method, and I feel that no algorithm is needed. Based on your suggestion, we have redescribed the mathematical formula to make it clearer.
Point5: The results and analysis section should be improved by adding more graphs, figures, tables, etc. to improve the quality of the manuscript.
Response 5: Thank you very much for your suggestions, we have made correction according to your comments. For example, different angles have been added to make the images clearer.
Point6: The conclusion section should be improved.
Response 6: Thank you for your comments, based on your suggestions, we have added research for future work to the conclusion and revised the conclusion for grammatical errors.
Point7: The authors should clearly describes the future scope of this work.
Response 7: Thank you for your advice, considering your suggestion, we have added the future scope of this work to the conclusion.
Point8: The authors should add 5-7 recent relevant references in the References section and cite them properly in the text.
Response 8: Thank you for your comments, according to your suggestion, we add five recent references to the references section. For example, the literature [15,16] and the literature [22-24].
Point9: The authors should go through the entire manuscript thoroughly and remove all the typo/grammatical errors from it.
Response 9: Thank you again for your valuable comments and suggestions, we invited a native English speaker to polish our manuscript, the entire manuscript grammatical errors has been modified.

Round 2
Reviewer 3 Report
There is no doubt that the quality of this manuscript has been improved after a round of revision. However, there may still be some problems and shortcomings. For example:
1. The symbol of the precision parameter is changed to α. However, the accuracy in the table and below is still R.
2. The selection of experimental factor level is still not explained thoroughly.
Of course, this is only for suggestion and discussion.
Author Response
Response to Reviewer 3 Comments
There is no doubt that the quality of this manuscript has been improved after a round of revision. However, there may still be some problems and shortcomings.
Point 1: The symbol of the precision parameter is changed to α. However, the accuracy in the table and below is still R.
Response 1: First, thank you very much for your reminding, we are very sorry for this. According to your advice, we have re-edited the formula to replace α with R.
Point 2: The selection of experimental factor level is still not explained thoroughly. Of course, this is only for suggestion and discussion.
Response 2: Thank you again for your valuable comments and suggestions and for your recognition of our revision work, your suggestions will greatly improve the quality of our manuscript. Based on your suggestions, we have reinterpreted the selection of the test factor level. (lines139-142)

Reviewer 5 Report
The authors attempt to solve the reviewer's comments. So, the manuscript should be accepted in present form.
Author Response
Response to Reviewer 5 Comments
The authors attempt to solve the reviewer's comments. So, the manuscript should be accepted in present form.
Response: Thank you again for your valuable comments and suggestions and for your recognition of our revision work, your suggestions will greatly improve the quality of our manuscript. On behalf of the co-authors, I would like to express my sincere gratitude to you.
